# Sustained Release of Tacrolimus Embedded in a Mixed Thermosensitive Hydrogel for Improving Functional Recovery of Injured Peripheral Nerves in Extremities

**DOI:** 10.3390/pharmaceutics15020508

**Published:** 2023-02-03

**Authors:** Aline Yen Ling Wang, Kuan-Hung Chen, Hsiu-Chao Lin, Charles Yuen Yung Loh, Yun-Ching Chang, Ana Elena Aviña, Chin-Ming Lee, I-Ming Chu, Fu-Chan Wei

**Affiliations:** 1Center for Vascularized Composite Allotransplantation, Chang Gung Memorial Hospital, Taoyuan 33305, Taiwan; 2Department of Physical Medicine & Rehabilitation, Chang Gung Memorial Hospital, Taoyuan 33305, Taiwan; 3College of Medicine, Chang Gung University, Taoyuan 33302, Taiwan; 4Department of Chemical Engineering, National Tsing Hua University, Hsinchu 30013, Taiwan; 5Department of Plastic Surgery, Addenbrooke’s Hospital, Hills Road, Cambridge CB2 0SP, UK; 6Department of Health Industry Technology Management, Chung Shan Medical University, Taichung 40201, Taiwan; 7Department of Medical Research, Chung Shan Medical University Hospital, Taichung 40201, Taiwan; 8International Ph.D. Program in Medicine, College of Medicine, Taipei Medical University, Taipei 11031, Taiwan; 9Department of Plastic Surgery, Chang Gung Memorial Hospital, Taoyuan 33305, Taiwan

**Keywords:** mixed thermosensitive hydrogel, tacrolimus, sustained release, nerve regeneration, sciatic nerve

## Abstract

Vascularized composite allotransplantation is an emerging strategy for the reconstruction of unique defects such as amputated limbs that cannot be repaired with autologous tissues. In order to ensure the function of transplanted limbs, the functional recovery of the anastomosed peripheral nerves must be confirmed. The immunosuppressive drug, tacrolimus, has been reported to promote nerve recovery in animal models. However, its repeated dosing comes with risks of systemic malignancies and opportunistic infections. Therefore, drug delivery approaches for locally sustained release can be designed to overcome this issue and reduce systemic complications. We developed a mixed thermosensitive hydrogel (poloxamer (PLX)-poly(l-alanine-lysine with Pluronic F-127) for the time-dependent sustained release of tacrolimus in our previous study. In this study, we demonstrated that the hydrogel drug degraded in a sustained manner and locally released tacrolimus in mice over one month without affecting the systemic immunity. The hydrogel drug significantly improved the functional recovery of injured sciatic nerves as assessed using five-toe spread and video gait analysis. Neuroregeneration was validated in hydrogel–drug-treated mice using axonal analysis. The hydrogel drug did not cause adverse effects in the mouse model during long-term follow-up. The local injection of encapsulated-tacrolimus mixed thermosensitive hydrogel accelerated peripheral nerve recovery without systemic adverse effects.

## 1. Introduction

Vascularized composite allotransplantation (VCA) can be used to reconstruct complex or large defects that cannot be repaired by conventional reconstruction surgery with autologous tissues, such as burned faces or amputated limbs. It restores not only the appearance of the defect, but also its function [1,2]. It can beautify a patient’s appearance and greatly improve their quality of life. In transplantation, nerve anastomosis needs to be performed between the allografts and recipients, meaning that nerve regeneration must be faced [3]. Limb or face transplantation requires the regeneration of peripheral nerves from the recipient body into the donor allograft. Peripheral nerves usually influence end-target organs and can cause denervation and loss of function if injured [3]. The issue, however, comes with the problem of axonal regeneration over a long distance. When this occurs, motor end-plate degeneration [4] follows, which prevents the axons from effectively restoring motor control over the end-target organ, which in VCA, is often a muscle. An example of this would be the restoration of intrinsic muscle function during hand transplantation. Often, extrinsic muscle function is restored, but the intrinsic muscles of the hand that are at a greater distance take longer for axonal regeneration. Coupled with issues of rejection, axonal regeneration is delayed further [1]. Compared to prosthetics, VCA allows patients the opportunity to restore tactility, aesthetics, sensation, and fine motor control in human tissues, which prosthetics cannot currently offer [5].

Peripheral nerve injuries are often severe and can cause severe disability with the loss of motor function. Movement recovery is highly time-sensitive and reinnervation loss can lead to the permanent loss of limb function, compounding already devastating social and economic consequences [6]. Nerve injuries far from end-target organs, such as muscles, need a much longer time to regenerate, resulting in the atrophy of motor end-plates and muscles [7]. New strategies are being developed in a great deal of research for the improvement of peripheral nerve regeneration, which ameliorates the outcomes following nerve injury [8,9,10,11,12].

Tacrolimus, also known as FK506, is an immunosuppressive drug widely used in transplantation to prevent allograft rejection. Tacrolimus interacts with immunophilin receptors, such as FK506-binding protein (FKBPs), to form an FKBP12–FK506 complex, and then inhibits calcineurin in the T cells [13]. Calcineurin is involved in T-cell activation by dephosphorylating the transcription factor nuclear factor of activated T-cells (NF-AT), which is associated with the signal pathways of T-cell activation and IL-2 production. Therefore, tacrolimus can suppress T-cell activation and proliferation by inhibiting calcineurin-mediated NF-AT dephosphorylation [14]. However, recent studies of tacrolimus have reported a secondary characteristic in promoting peripheral nerve recovery in animal models such as rats, mice, rabbits, and primates [15,16,17,18,19,20,21,22,23,24,25,26,27,28,29,30]. Functional recovery and the maintenance of muscle bulk have been reported, as well as significant improvements in the speed of peripheral nerve regeneration after tacrolimus systemic administration [21,31].

However, systemically delivered tacrolimus is not used clinically for peripheral nerve regeneration due to concerns about its systemic immunosuppressive effects, which result in undesirable and erratic consequences, including life-threatening opportunistic infection and malignant lesions [32,33,34,35,36,37]. Additionally, compliance or regular use of the drug can be an issue, with daily administration needed to maintain drug concentrations. Therefore, local delivery of the drug may reduce the administered dosage and systemic complications, while maintaining optimal concentrations of the drug at the site of the peripheral nerve injury. The development of a novel drug-binding bioengineered construct that both effectively holds tacrolimus and provides a time-dependent, persistent release of the drug into the local site of the peripheral nerve injury would be a revolutionary form of treatment. This would enhance the reliability of drug delivery and also ensure adequate concentrations at the site of injury, thereby promoting peripheral nerve recovery and minimizing systemic effects.

Hydrogels, which can transport therapeutic drugs in a highly hydrated depot-like form, are made up of crosslinked polymeric networks [38,39,40]. A thermosensitive hydrogel has a hydrophobic group in its structure in addition to the hydrophilic group. Both the hydrophilic and hydrophobic groups compete with each other. When the temperature is low, hydrogen bonding occurs between the water molecules and the hydrophilic groups. Thus, the polymer chain can be completely stretched in water after absorbing water to form a uniform solution. When a certain temperature is reached, the hydrogen bonding force between the molecules becomes weaker. Thus, the hydrophobic group aggregates in water and causes the polymer solution to change from a liquid state to a colloidal state. In our previous study, in order to deliver therapeutic drugs at a specific site in vivo via gelling at 37 °C, a thermosensitive hydrogel composed of poloxamer and poly(l-alanine) with l-lysine segments at both ends (P–Lys–Ala–PLX) was developed and synthesized [41]. The average molecular weight (Mw) of this triblock copolymer is 3817 Da and the molar mass polydispersity is 1.51. It can transport tacrolimus in a gelled form with properties of biodegradability, biocompatibility, and low gelling concentrations from 3–7 wt%. Additionally, in order to promote tacrolimus release at effective concentrations and the continued release for around one month, Pluronic F-127 was added to the thermosensitive hydrogel to form a mixed thermosensitive hydrogel system. Pluronic F-127 is a commercially available and FDA-approved fast-degrading hydrogel, reported to regulate the release rate of encapsulated drugs [42]. As 5 wt% of P–Lys–Ala–PLX combined with 1 wt% of Pluronic F-127 had the highest encapsulation efficiency of tacrolimus, this formulation was used to encapsulate and release tacrolimus over a certain period of time in our studies. We found that Pluronic F-127 help the mixed hydrogel to release the drug through the acceleration of hydrogel degradation at 37 °C compared with P–Lys–Ala–PLX hydrogel only. In this study, we used the mixed thermosensitive hydrogel system carrying tacrolimus to evaluate its efficacy in terms of the functional recovery and neuroregeneration of peripheral nerve injury in a mouse transected sciatic nerve model.

## 2. Materials and Methods

### 2.1. Mice

C57BL/6J mice (8–10 weeks old) purchased from the National Laboratory Animal Center, Taiwan, were maintained in the AAALAC accredited animal center of the Chang Gung Memorial Hospital (CGMH) and used for all of the experiments in the study. The animal protocols were conducted in compliance with the Guide for the Care and Use of Laboratory Animals (Chang Gung Memorial Hospital Animal Research Guidelines). The mouse nerve transection protocol was approved by the Committee on the Ethics of Animal Experiments of the CGMH in Taiwan and the Institutional Animal Care and Use Committees (IACUC) of CGMH in Taiwan under permit numbers IACUC 2019120201, IACUC 2020121608, IACUC 2021031503, and IACUC 2021091405.

### 2.2. Tacrolimus-Laden Mixed Thermosensitive Hydrogel System

The hydrogel–tacrolimus delivery system was engineered and provided by Prof. I-Ming Chu. Poloxamer was used as the main polymer chain and was copolymerized with l-alanine and l-Lysine s to form the copolymer. Pluronic^®^ is a commercially available, FDA-approved biocompatible polymer. The system was made with 5 wt% PLX-poly(l-alanine-lysine) (P–Lys–Ala–PLX) and 1 wt% Pluronic F-127 in a mixed form to release tacrolimus at a steady rate and thus ensure effective drug levels over a certain period of time. Details of the synthesis, biocompatibility, and tacrolimus-release rate of this system are described in our previous study [41]. Tacrolimus-carrying hydrogels were prepared in 0.1–2 mg/mL formulae in this study.

### 2.3. Flow Cytometry

Immune cells from lymphoid organs, such as blood, spleen, and lymph nodes, were stained with cell surface antibodies against CD3, CD4, CD8, CD19, CD11c, or NK1.1, purchased from eBioscience (San Diego, CA, USA). An FACSCanto II flow cytometer (BD Biosciences, San Jose, CA, USA) was used to examine the cell surface antibody-stained immune cells over a 30 min incubation at 4 °C. For CD4^+^CD25^+^FoxP3^+^ regulatory T cell (Tregs) staining, the immune cells were intracellularly stained with FoxP3-PerCP (eBioscience, San Diego, CA, USA) after cell surface staining of CD4-APC and CD25-PE and fixation/permeabilization.

### 2.4. Cytometric Bead Array Assay

The concentrations of pro-inflammatory cytokines, such as interleukin (IL)-12p70, IL-6, TNF-α, and interferon (IFN)-γ, anti-inflammatory IL-10 cytokine, and chemokine monocyte chemoattractant protein-1 (MCP-1), were examined via a cytometric bead array (CBA) assay (BD, Franklin Lakes, NJ, USA) using flow cytometry. The various cytokine concentrations (pg/mL) were calculated according to the standard curve using the FCAP Array multiplex analysis software.

### 2.5. Quantification of Tacrolimus Concentration in Tissues

To analyze the tacrolimus-release pharmacokinetics, we took samples of blood plasma and 100 mg samples of muscle and nerve tissue near the hydrogel injection site (Tissue-S) and at the opposed hindlimb area (Tissue-O) at the designated time points after hydrogel–tacrolimus injection. Then, 400 μL of T-PER Tissue Protein Extraction Reagent (Thermo Fisher Scientific, Carlsbad, CA, USA) was added to each of the various tissue samples and homogenized using a Precellys homogenizer with the addition of adding ceramic beads (3 mm). The homogenates were centrifuged at 15,000 rpm for 15 min at 4 °C and the supernatants were evaluated using FK506 ELISA. Consequently, the tacrolimus concentration in either the plasma or tissue was analyzed using an Abona FK506 ELISA kit (Abnova, Taiwan), according to the manufacturer’s instructions. A SpectraMax reader was used to read the absorbance at 495 nm.

### 2.6. Mouse Sciatic Nerve Transection Model

Inhaled isoflurane was used for anesthesia throughout the surgery. Pre-operation preparation included shaving the posterior surface of the right hindlimb using mouse clippers and sterilizing the skin with 75% ethanol. A heating pad was used to maintain the animal temperature at 37 °C. The sciatic nerve was exposed by opening the fascial plane between the anterior head of the biceps femoris and the gluteus maximus. The sharp division of the sciatic nerve was conducted under a dissecting microscope with microsurgical scissors at the mid-thigh level, 5 mm proximal to the trifurcation of the sciatic nerve. Subsequent coaptation in the epineural layer was performed with four evenly spaced sutures using 10/0 nylon sutures. Finally, the skin incision was closed using 5/0 vicryl sutures.

### 2.7. Topical Administration of Hydrogel Control and Hydrogel-Tacrolimus

A suspension of 0.1, 0.2, 0.5, 1, or 2 mg/mL of tacrolimus-carrying hydrogel was injected into the subcutaneous space near the injured nerves following the transection and anastomosis of the sciatic nerves. The same volume (0.1 mL) of hydrogel was subcutaneously injected into the control mice after the sciatic nerve transection was performed. In this study, a total of sixty mice were used, as shown in Figure 1, Figure 2 and Figure 3, and the hydrogel and Hydrogel-Tac (2 mg/mL) groups had 30 mice, respectively, for different time points. Additionally, a total of forty-seven mice were used for the evaluation of functional recovery, as shown in Figure 4. The hydrogel, Hydrogel-Tac (0.1 mg/mL), Hydrogel-Tac (0.2 mg/mL), Hydrogel-Tac (0.5 mg/mL), Hydrogel-Tac (1 mg/mL), and Hydrogel-Tac (2 mg/mL) groups had six, five, five, eight, five, and six mice, respectively, for the measurements of the five-toe spread in the hydrogel and Hydrogel-Tac groups following sciatic nerve transection up to postoperative day (POD) 225, as shown in Figure 4B. These mice were also used for the video gait analysis (Figure 5), axonal analysis (Figure 6), and safety profiles (Figure 7). The hydrogel and systemic Tac groups contained six mice, respectively, for the measurements of the five-toe spread following sciatic nerve transection up to POD 99 in Figure 4C.

### 2.8. Functional Recovery by Five-Toe Spread Analysis

Five-toe spread analysis represents the intrinsic muscle recovery and indicates the signal transmission recovery after nerve transection [9,11,12]. In the present study, the distance of the five-toe spread was estimated in each mouse before the nerve transection. The measurements of the five-toe spread were recorded at different time points and the average score of the four measurements was taken. The cumulative graph indicates the percentage of the postoperative five-toe spread distance divided by pre-transection (Figure 4).

### 2.9. Functional Recovery by Video Gait Angle Analysis

The ankle angles while walking were also used to monitor sciatic nerve recovery after nerve transection. The sciatic nerve innervates various leg muscles and maintains these muscles, permitting normal gait. Ankle movements are therefore improved during mouse nerve recovery. Studies have shown that ankle angle is a good and noninvasive indicator of the isometric force produced by the leg muscles in various gait phases [9,11,12]. The mouse gait cycle is separated into four stages: foot on the ground, midstance phase, toe-off phase, and mid-swing phase. During the toe-off phase, a larger ankle angle has been reported to be positively associated with the functional recovery of sciatic nerves. The gait motion during walking was recorded using a 60 Hz digital camera. The average of four measurements was taken for the ankle angles during walking.

### 2.10. Nerve Regeneration by the Analysis of Toluidine Blue Staining

Biopsy of the sciatic nerves in the hydrogel and encapsulated-tacrolimus hydrogel (0.1, 0.2, 0.5 mg/mL) groups was performed on POD 225. The nerve tissues of both groups near the transection area—defined as the proximal sections—and nerve tissues at a fixed distance of 5 mm away from the transection site—defined as the distal sections—were examined. The nerve tissues were stained using toluidine blue to better analyze the degree of axonal growth under a cross-sectional view [9,11,12]. The nerve fiber was then cut into 60 nm sections and fixed with lead citrate and uranyl acetate. Photographs of the cross-section were taken under a light microscope and the Image-Pro 2D/3D/4D image analysis software was used to precisely measure the axon diameter, nerve fiber diameter, and myelin thickness. The G ratio is the ratio of the axon diameter to the nerve fiber diameter and indicates the quality of the nerve signal conduction.

### 2.11. Histological Examination by Hematoxylin and Eosin Staining

Hematoxylin and eosin (H&E) staining was used to examine various organ tissues from different hydrogel-treated mice, which were collected on day 289 after nerve transection. The tissues were preserved in 4% formaldehyde before staining. The tissue architecture and pathological abnormalities were examined via microscope observation to evaluate the safety profiles of the hydrogel and encapsulated-tacrolimus hydrogel treatments.

### 2.12. Statistical Analysis

Mean ± SD was used to represent the statistical data. One-way ANOVA and post hoc analysis using Tukey’s multiple comparisons test were used for the analysis of the variation in the five-toe spread and ankle angle. For continuous variables, the Student’s *t*-test was used. The GraphPad Prism 6 software (San Diego, CA, USA) was used for statistical analysis. The definition of statistical significance was *p* < 0.05.

## 3. Results

### 3.1. Hydrogel Degradation Kinetics

In our previous study, a mixed thermosensitive hydrogel (poloxamer (PLX)-poly(l-alanine-lysine), P–Lys–Ala–PLX) was developed for the improvement of the encapsulation efficiency by the hydrogel of a hydrophobic drug without a co-solvent, and it displayed a time-dependent sustained release of tacrolimus [41]. Moreover, the thermosensitive hydrogel was designed to present a liquid form at 4 °C and a solid form at 37 °C to facilitate the fixation of the hydrogel drug in one place in an animal’s body for the sustained release of the drug. We currently have a poor understating of this tacrolimus-bound hydrogel construct in vivo; hence, the degradation kinetics of the new hydrogel–drug system in the mouse model were first examined. Two milligrams of tacrolimus was incorporated into 1 mL of polypeptide thermosensitive hydrogel, forming 2 mg/mL of encapsulated-tacrolimus hydrogel (Hydrogel-Tac). To investigate the effect of Hydrogel-Tac on the functional recovery of the damaged sciatic nerve, 0.1 mL of 2 mg/mL Hydrogel-Tac or hydrogel was injected into the hindlimb subcutaneous space near the transected sciatic nerve, as depicted in Figure 1A. The injected hydrogels were harvested and weighed at different time points. The size and volume of both hydrogels were observed to gradually degrade, disappearing around day 43 (Figure 1B). The measurements of hydrogels in both groups were statistically calculated, as presented in Figure 1C. Both of the hydrogels subcutaneously injected into the mouse hindlimb were degraded to around 50% on day 14 and totally degraded on day 43. The speed and degree of degradation were similar between the hydrogel and Hydrogel-Tac groups. These results indicate that the hydrogel and tacrolimus-laden hydrogel were gradually degraded during a period of time in the mouse model.

### 3.2. In Vivo Effects of Hydrogel-Tac Local Administration on the Systemic Immunomodulation

Tacrolimus has immunosuppressive functions in addition to promoting nerve regeneration. The local effect of Hydrogel-Tac is designed to result in minimal systemic effects on the recipient immune system while promoting axonal regeneration. The mouse hindlimb subcutaneous space was injected with 0.1 mL of hydrogel or Hydrogel-Tac (2 mg/mL). At different time points, lymphoid organs, such as blood, spleen, and lymph nodes (LNs), were harvested and analyzed using flow cytometry. LNs from near the anastomosed sciatic nerve (LN-S) and LNs from the opposing hindlimb area (LN-O) were both obtained for examination. We found that the compositions of lymphocytes, monocytes, and granulocytes were similar between the hydrogel and Hydrogel-Tac groups (Figure 2A). Further analysis of the lymphocyte subsets in lymphoid organs was performed at different time points. The Hydrogel-Tac group displayed similar populations of CD3^+^, CD4^+^, and CD8^+^ T cells compared with the hydrogel group (Figure 2B). Innate immunity is represented by natural killer cells (NK1.1^+^, NKs), which secrete inflammatory interferon-γ (INF-γ); populations of these cells were also similar between both groups, as were those of natural killer T cells (NK1.1^+^CD3^+^, NKTs) (Figure 2C). NKT cells can produce large amounts of interleukin (IL)-4 and granulocyte-macrophage colony-stimulating factor in addition to interferon-γ after activation. The CD19^+^ B cells and CD11c^+^ dendritic cells (DCs), which are crucial for allorecognition and T-cell activation, showed similarities between both groups (Figure 2D). A similar population of anti-inflammatory regulatory T cells (CD4^+^CD25^+^FoxP3^+^ Tregs) was observed in both groups (Figure 2E). These results suggest that the single local injection of Hydrogel-Tac had minimal effects on the systemic immune cell profiles, including inflammatory and anti-inflammatory cell subsets.

To further evaluate the effects of Hydrogel-Tac on the pro-inflammatory and anti-inflammatory cytokine profiles in blood sera, we used a CBA assay to examine cytokine production using flow cytometry. Analyzed cytokines included pro-inflammatory tumor necrosis factor-alpha (TNF-α), IL-12, IFN-γ, monocyte chemoattractant protein-1 (MCP-1), IL-6, and anti-inflammatory IL-10. The levels of pro-inflammatory IL-12 and IL-6 cytokines appeared to be higher on day three in the Hydrogel-Tac-treated mice. As the study progressed, the cytokines showed similar overall levels between the hydrogel- and Hydrogel-Tac-treated mice (Appendix A). These results indicate that a single topical administration of Hydrogel-Tac had minimal effects on the systemic cytokine profiles, including pro-inflammatory and anti-inflammatory cytokines.

### 3.3. In Vivo Sustained-Release Pharmacokinetics in Hydrogel-Tac Topical Application

To understand the drug-release kinetics of the locally administered encapsulated-tacrolimus hydrogel in vivo, the examination of the tacrolimus concentration was conducted in both the hydrogel and Hydrogel-Tac (2 mg/mL) groups. Systemic plasma, hindlimb muscle, and sciatic nerve tissues from near the hydrogel-injected site (Tissue-S), and muscle and nerve tissues from the opposing hindlimb area (Tissue-O), were harvested on days 0, 1, 2, 3, 7, 14, 28, 35, and 43, and then analyzed using a tacrolimus ELISA assay. Day 0 indicates samples collected from the naïve C57BL/6 mice without any hydrogel injection. As shown in Figure 3A, the tacrolimus concentrations in the plasma of both the hydrogel groups during the 43 day period following one local administration were similar, at approximately 1 ng/mL, which is generally a background level suggesting the minimal systemic release of the local Hydrogel-Tac injection. On the other hand, the tacrolimus concentration in the Tissue-S of the Hydrogel-Tac group was approximately 300 ng/g on day one and showed continuous release until day 43. The tacrolimus concentration above 24 ng/g in the Hydrogel-Tac group was sustained to day 14 and gradually decreased to a level similar to that of the hydrogel control group by around day 43 (Figure 3B). The Tissue-O samples of both groups displayed similar low concentrations of tacrolimus, usually a background level, which were similar to the plasma results (Figure 3B). These results suggest that one local injection of 0.1 mL of encapsulated-tacrolimus hydrogel (2 mg/mL) exhibited local sustained-release pharmacokinetics with minimal systemic effects.

### 3.4. Efficacy of Hydrogel-Tac Topical Administration for Injured Sciatic Nerve Recovery

Five-toe spread and video gait analysis were used to evaluate the functional recovery of injured sciatic nerves [9,11,12] after a single topical administration of Hydrogel-Tac. In order to assess the efficacy of the Hydrogel-Tac topical treatment on sciatic nerve recovery, we recorded the measurements of five-toe spread weekly following nerve transection. As shown in Figure 4A, the full abduction of the five toes was observed in the normal mice without nerve transection, whereas the toes were barely open or were even shriveled in the hydrogel-treated mice on POD94 after nerve transection. The nerve-injured mice treated with different dosages of Hydrogel-Tac displayed a dose-dependent improvement of their five-toe spread, particularly those in the 0.5 mg/mL Hydrogel-Tac group, which showed the largest abduction of the five-toe spread, similar to the normal mice.

The percentages of the postoperative five-toe spread distances in the presence of the different hydrogel treatments divided by the pre-transection measurement were used to evaluate the efficacy of Hydrogel-Tac on sciatic nerve recovery, as illustrated in Figure 4B. The hydrogel group used as the negative control showed spontaneous nerve recovery after nerve transection and reached 59.2% five-toe spread on POD 60. The group percentage of the five-toe spread then gradually decreased to the initial percentage. Clawing of the toes and the fixed flexion contracture of the toe joints were noticed in the hydrogel group, as shown in Figure 4A. However, most dosages of Hydrogel-Tac (0.2–1 mg/mL) were associated with significantly accelerated recovery in the first 30 days and peak five-toe spread percentages were reached at around 30–95 days. As shown in Figure 4B, the mice dosed with Hydrogel-Tac (0.1–0.5 mg/mL) exhibited functional improvement in a dose-dependent manner. The 0.5 mg/mL group showed the highest and most persistent improvement up to POD 225, whereas the 0.1 mg/mL group showed poor improvement similar to the hydrogel group. Although the 1 mg/mL group reached around 80% improvement on POD 30, similarly to the 0.5 mg/mL group, the five-toe spread percentages gradually reduced to 63.4% and 79.3% on POD 225 in the 1 and 0.5 mg/mL groups, respectively. Interestingly, the 2 mg/mL group only reached 72.6% improvement on POD30, and this gradually decreased to 54.8% on POD 225. These results indicate that a single topical injection of 0.5 mg/mL of Hydrogel-Tac was the optimal tacrolimus concentration for the long-term maintenance of the best functional recovery. An optimal dosage of topically administered Hydrogel-Tac offers the potential for faster reinnervation, resulting in accelerated sciatic nerve recovery and the maintenance of five-toe abduction.

The systemic administration of tacrolimus has been reported to accelerate nerve functional recovery [18]; hence, the daily intraperitoneal injection of tacrolimus (2 mg/mL) was used as a positive control until the end of the observation day POD 99. As shown in Figure 4C, the systemic tacrolimus group displayed significant improvements in nerve functional recovery on POD 37, 51, and 99 compared with the hydrogel local injection group, according to the five-toe spread analyses. The systemic tacrolimus group reached a maximum improvement percentage of around 75% on POD 37; subsequently, a decay of the five-toe spread distance was noticed over long-term observation.

The functional recovery of the sciatic nerve in mice was also assessed and monitored using video gait analysis, which provided information on muscle force and muscle bulk maintenance. There are four stages in the mouse gait cycle, as previously described in the Methods. By evaluating the gait angles of the toe-off phase, we were able to anticipate muscle force generation; larger angles in this phase indicate better nerve recovery and muscle bulk maintenance. The gait angle became smaller if the muscle was denervated and eventual limb contracture occurred. As a single topical injection of 0.5 mg/mL of Hydrogel-Tac was associated with the long-term maintenance of the best functional recovery according to the five-toe spread analysis (Figure 4), video gait analysis was further used to confirm sciatic nerve recovery. As shown in Figure 5A, the representative photographs of gait angles in the Hydrogel-Tac group on POD 122 showed large angles, similar to those seen in the normal group without nerve transection. Contracture and hindlimb bending were noticed in the hydrogel group, suggesting muscle wasting and poor nerve recovery after nerve transection. On POD 1, the gait angles of both the hydrogel and Hydrogel-Tac groups showed a sharp decrease in the toe-off phase, indicating that these mice indeed received the nerve-transection operation. Compared with the hydrogel group, the Hydrogel-Tac groups displayed significant increases in angles at faster rates (Figure 5B). Significant differences in the gait angles were documented on POD 30, 95, 122, 156, and 225 (*p* = 0.0095, 0.027, 0.0121, 0.0011, and <0.0001, respectively; unpaired *t*-test). The normal control mice without sciatic nerve transection were also evaluated; their average angle was 125.6 ± 1.392 degrees. These results demonstrated that one dose of topically administered encapsulated-tacrolimus hydrogel near the nerve transection site promoted and maintained the long-term functional recovery of sciatic nerves, as shown by the analyses of both the five-toe spread and gait angles.

### 3.5. Efficacy of Hydrogel-Tac Topical Administration for Axonal Regeneration

To further understand the effects of encapsulated-tacrolimus hydrogel on sciatic nerve regeneration, myelin sheath staining with toluidine blue was performed and analyzed using light microscopy. Proximal and distal biopsies of the sciatic nerves were, respectively, performed near the transected area and at a fixed distance from the point of transection. As shown in Figure 6A, the representative photographs of the myelin sheath staining showed the myelin thickness, nerve fiber diameter, and axon diameter, and were measured with the Image-Pro 2D/3D/4D image analysis software. Representative cross-sections of sciatic nerves in the hydrogel and Hydrogel-Tac (0.5 mg/mL) groups on POD 225 are shown in Figure 6B. In the proximal sections, the different concentrations of Hydrogel-Tac displayed similar axon counts and axon areas compared with the hydrogel control group. However, in the distal section, significantly greater numbers of axons were noticed in the 0.2 and 0.5mg/mL Hydrogel-Tac groups compared with the hydrogel group (Figure 6C). The 0.5 mg/mL Hydrogel-Tac group also had a larger total axonal area in the distal section when compared to the hydrogel group. To explore the effects of Hydrogel-Tac on the speed of the nerve fiber conduction and myelination, the g ratio was calculated (axon diameter to nerve fiber diameter). Its range generally varies between 0.5 and 0.8 in all nerve fibers [9,11,43,44,45,46,47,48]. In this study, the g ratios were a little higher, suggesting relatively thinner myelin and small myelinated axons. In the proximal section, both groups revealed similar g ratios (0.8221 vs. 0.8721), suggesting similar nerve conduction. In the distal section, the Hydrogel-Tac group had a slightly higher g ratio than the hydrogel group (0.898 vs. 0.6877) (Figure 6D). These results indicate that one dose of topically administered encapsulated-tacrolimus hydrogel (0.5 mg/mL) accelerated neuroregeneration, including the axonal number and area, resulting in the functional recovery of the sciatic nerves.

### 3.6. Long-Term Systemic Effects of Local Hydrogel-Tac Treatment

To evaluate the effects of Hydrogel-Tac on systemic organ inflammation in a long-term follow-up, lymphocyte infiltration into various organs was verified via histological H&E staining. Both the macroscopic and microscopic examination were used for systemic evaluation on POD 289 after the nerve transection. Representative photographs of the nerve-sutured site and the surrounding tissue architecture were documented for each group (Figure 7A). There were no adverse effects observed in the nerve-sutured site following hydrogel injection. These hydrogel-treated mice were healthy and active before the harvesting of the biopsy samples. Heart, liver, intestine, kidney, and spleen biopsies were taken at random sites and stained with H&E (Figure 7B). Additionally, there were no significant pathological findings, such as tissue necrosis, hematoma, or inflammation, upon microscopic examination of these organs in any of the hydrogel groups. These results indicate that neither hydrogel nor Hydrogel-Tac administration resulted in noticeable adverse effects based on macroscopic and microscopic examination in a long-term follow-up, suggesting minimal toxicity for the host.

## 4. Discussion

As a result of the properties of tacrolimus, such as its high hydrophobicity and low aqueous solubility, the formulation of tacrolimus injection is usually complicated, and co-solvents, such as ethanol or dimethyl sulfoxide (DMSO), are often used. However, co-solvents are not only deleterious to cellular organisms, but also increase the risks associated with drug use in patients. In our previous study, mixed thermosensitive hydrogels composed of poloxamer (PLX)-poly(l-alanine-lysine) and Pluronic F-127 were designed and demonstrated to deliver the hydrophobic tacrolimus drug without co-solvents [41]. The encapsulated-tacrolimus hydrogel possesses features of biodegradability, biocompatibility, thermosensitive gelling at 37 °C, and no requirement for additional enzymes to release the tacrolimus. In this study, we used a hydrogel–tacrolimus formulation (Hydrogel-Tac) to treat transected sciatic nerves with one dose in a mouse nerve model. We found that the optimal dosage of Hydrogel-Tac was around 0.5 mg/mL with an injection of 0.1 mL, which produced the best functional recovery using a five-toe spread analysis (Figure 4). In the video gait analysis, 0.5 mg/mL of Hydrogel-Tac significantly promoted functional nerve recovery, with the largest angle gait after the long-term follow-up (Figure 5). We further used toluidine blue to stain the nerve myelin for the examination of sciatic nerve regeneration. In the distal sciatic nerve sections of the Hydrogel-Tac (0.5 mg/mL) group, the total axonal counts and total axon areas were significantly increased compared with the hydrogel group, but they were not increased in the proximal sections. The Hydrogel-Tac group also had a better nerve conduction rate than the hydrogel group, as analyzed by the g ratio (Figure 6). As for the safety profile of encapsulated-tacrolimus hydrogel, no adverse effects and no evidence of excessive inflammatory cell infiltration into various organs were observed 289 days postoperatively in either the hydrogel or Hydrogel-Tac groups (Figure 7). In the mouse model, although both hydrogels displayed similar degradation kinetics and immune cell profiles within 43 days, the sustained presence of tacrolimus was limited to the injected tissues of the Hydrogel-Tac group (Figure 1, Figure 2 and Figure 3). These results indicate that topically applied hydrogel-encapsulated tacrolimus achieved sustained release, promoting sciatic nerve regeneration and functional recovery through a local therapeutic effect.

Various animal studies on nerve regeneration and functional recovery after nerve crush or transection have utilized different dosages of tacrolimus with different administration routes. A study reported that the systemic administration, by subcutaneous injection, of tacrolimus can promote axonal regeneration and functional recovery in a dose-dependent manner after sciatic nerve crush injury. A subcutaneous dose of 5 mg/kg daily had the greatest efficiency for rat neuroregeneration [49]. Yang et al. reported that a subcutaneous dose of tacrolimus greater than 1 mg/kg daily was needed to accelerate nerve regeneration in rats [17]. The evidence of oral administration of tacrolimus for rat neuroregeneration showed that a daily dose of 15 mg/kg was the most effective [31]. A mouse study reported that a subcutaneous dose of 5 mg/kg daily was the most effective for axonal regeneration following peripheral nerve crush injury, whereas a daily dose of 10 mg/kg produced results similar to those seen in the control group [16]. In our study, only one subcutaneous injection of encapsulated-tacrolimus hydrogel efficiently promoted functional recovery following sciatic nerve injury, as assessed using five-toe spread analysis (Figure 4). The optimal dosage of encapsulated-tacrolimus hydrogel for functional recovery was 0.5 mg/mL. The efficiency of neuroregeneration displayed a dose-dependent trend between 0.1 and 0.5 mg/mL, whereas the effect decreased with more than 0.5 mg/mL. Moreover, the systemic intraperitoneal administration of tacrolimus through daily injection of 2 mg/kg up to the observation end date was associated with a similar recovery to one dose of Hydrogel-Tac (0.2 mg/mL) (Figure 4C). Our results indicate that one topical administration of encapsulated-tacrolimus hydrogel can greatly reduce the therapeutic dosage of tacrolimus and avoid repeated systemic administration. This could further decrease the potential side effects of systemic tacrolimus administration, such as body weight loss, diarrhea, hyperglycemia, tremors, asthenia, paresthesias, pain, headaches, seizures, nephrotoxicity, central nervous system toxicity, and tumor risk [50,51]. Therefore, tacrolimus drug delivery can be designed for local sustained release to overcome these issues and reduce the systemic complications, while maintaining optimal concentrations.

Vascularized composite allotransplantation is an emerging reconstructive strategy for unique defects that cannot be repaired with conventional autologous tissues, such as amputated limbs or severely burned faces. In order for transplanted limbs to be functional, functional recovery of the peripheral nerves must be ensured. The recovery of movement and sensation is normally impaired and slow. The loss of reinnervation results in the permanent loss of function of a limb; thus, promoting the improvement of the motor and sensory function of the transplanted limb through the recovery of peripheral nerves is paramount. Therefore, in VCA, nerve anastomosis needs to be performed in addition to vascular coaptation between the allografts and recipients, meaning that two challenges must be faced: allograft rejection [52,53,54] and nerve regeneration [3]. Encapsulated-tacrolimus hydrogel, with its two properties of inhibiting allograft rejection and promoting nerve regeneration, therefore holds promise for allogeneic limb transplantation. In this study, we designed a hydrogel for timed drug release over one month to improve functional recovery of the syngeneic injured sciatic nerve. In order to treat allogeneic limb transplantation, the formulation of the hydrogel drug need to be adjusted. The drug release time of the hydrogel can be prolonged by increasing the wt% if the polymer structure or molecular weight is the same. If the wt% is the same, the amount of poly (Alaina) monomer in PLX–Ala–Lys can be increased; that is, the molecular weight of the polymer can be increased to prolong the drug release time and hydrogel degradation time.

## 5. Conclusions

Tacrolimus is an immunosuppressive drug used for long-term immunosuppression in transplant patients to prevent transplant rejection through the inhibition of calcineurin. It also has the potential to improve peripheral nerve recovery in animal models. The highly hydrophobic tacrolimus can be embedded in a mixed thermosensitive hydrogel, which can promote functional recovery through sustained release to the injured peripheral nerve. Encapsulated-tacrolimus hydrogel, with its two characteristics of inhibiting allograft rejection and promoting nerve regeneration, is suitable for allogeneic limb transplantation because limb allotransplantation faces two challenges: allograft rejection and nerve regeneration. The newly designed mixed thermosensitive hydrogel formulation holds promise for tacrolimus delivery in vitro and in vivo, as well as for the delivery of other highly hydrophobic drugs.

## 6. Patents

As demonstrated in this study, a mixed thermosensitive hydrogel system (P–Lys–Ala–PLX combined with Pluronic F-127) was applied to promote peripheral axonal regeneration and functional recovery in a mouse model. The patent (TWI649096B) for the hydrogel application has been approved in Taiwan, reporting the improvement of the encapsulation efficiency of hydrogel for a hydrophobic drug without a co-solvent [55]. Hydrophobic drugs, such as tacrolimus, can be released in a sustained and continuous manner by this drug delivery system, providing a long-term therapeutic effect. This hydrogel does not require additional enzymes to release tacrolimus. Moreover, the gel is formed when the temperature is greater than 4 °C and less than 37 °C. This indicates that a hydrogel drug will be a liquid at 4 °C when prepared and a solid at 37 °C when injected into animals. This temperature-dependent design facilitates the fixation of the hydrogel in one place in an animal’s body for the sustained release of the drug. We hope that the mixed thermosensitive hydrogel will encourage the further application of other hydrophobic drugs in animal models, which will have applicability in many clinical scenarios.

## Figures and Tables

**Figure 1 pharmaceutics-15-00508-f001:**
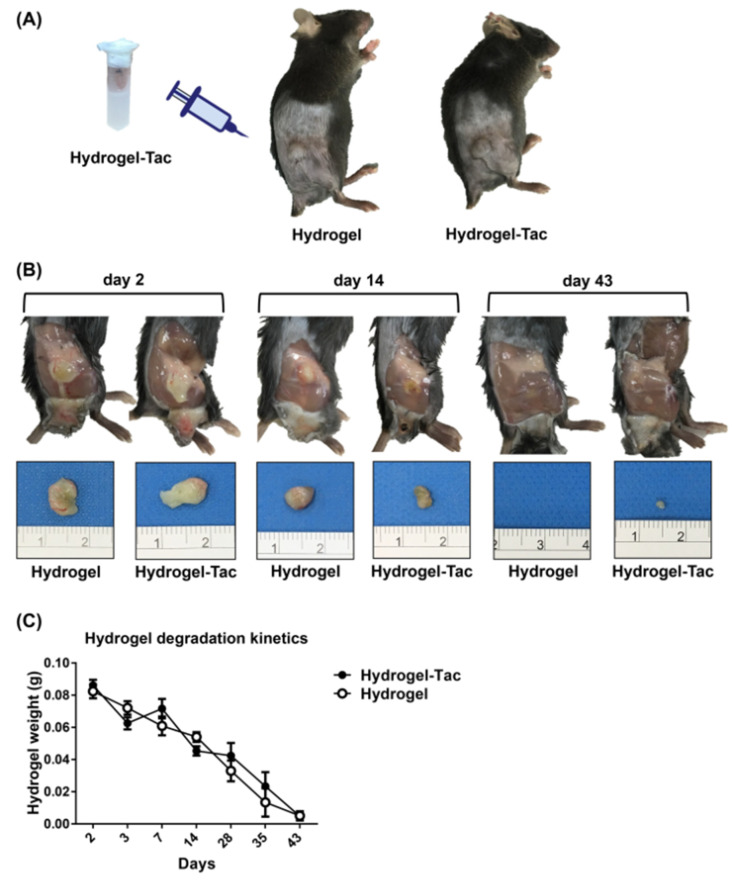
The degradation kinetics of hydrogel in the mouse model. (**A**) 2 mg of tacrolimus was encapsulated in 1 mL of polypeptide thermosensitive hydrogel, forming 2 mg/mL of Hydrogel-Tac. Subsequently, 0.1 mL of Hydrogel-Tac or hydrogel was injected into the subcutaneous space of the hindlimb, which was near the transected sciatic nerve. (**B**,**C**) The hydrogels with or without tacrolimus were harvested at different time points from the mice injected with one dose of 0.1 mL of hydrogel or Hydrogel-Tac (2 mg/mL) and then weighed. The smallest unit of measurement was a millimeter. Each group had three mice. Mean ± SD was used to represent the statistical data.

**Figure 2 pharmaceutics-15-00508-f002:**
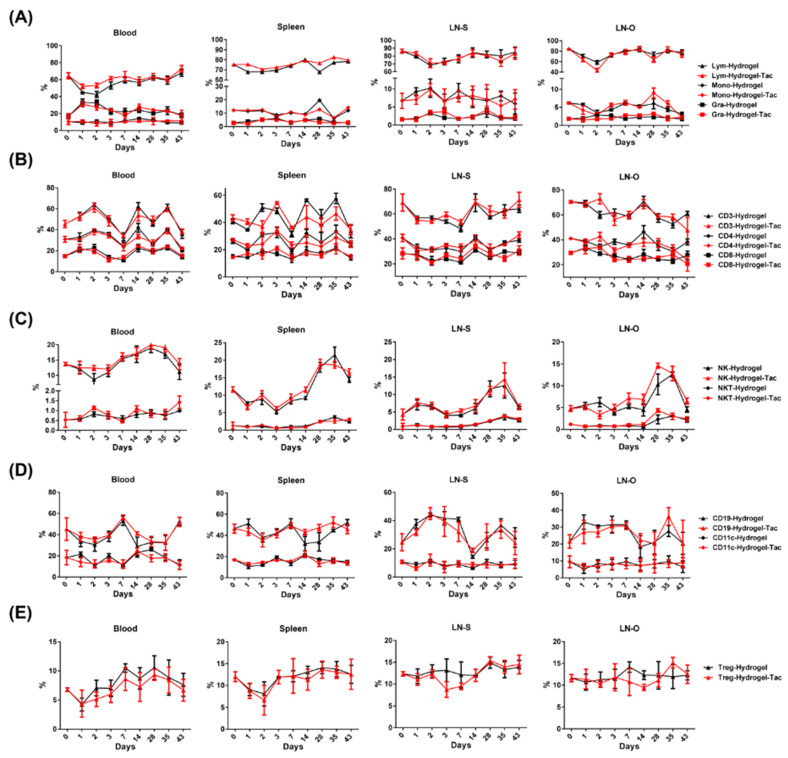
The effects of Hydrogel-Tac local injection on the systemic immune system. At different time points, various lymphoid organs, such as blood, spleen, and LNs, were harvested from the mice treated with one subcutaneous injection of 0.1 mL of hydrogel or Hydrogel-Tac (2 mg/mL) and then analyzed by flow cytometry. LN-S refers to the LNs on the same side as the hydrogel injection and near the right hindlimb. LN-O refers to the LNs on the opposite side to the hydrogel injection and near the left hindlimb. (**A**) Lymphocyte, monocyte, and granulocyte subsets were analyzed between the hydrogel and Hydrogel-Tac groups. (**B**) CD3^+^, CD4^+^, and CD8^+^ T-cell subsets were examined between both groups. (**C**) Natural killer (NK1.1^+^, NK) and natural killer T (NK1.1^+^CD3^+^, NKT) cell subsets were determined between both groups. (**D**) CD19^+^ B and CD11c^+^ dendritic cell subsets were verified between both groups. (**E**) CD4^+^CD25^+^FoxP3^+^ regulatory T (Treg) cell subset was evaluated between both groups. Each group had three mice. Mean ± SD was used to represent the statistical data.

**Figure 3 pharmaceutics-15-00508-f003:**
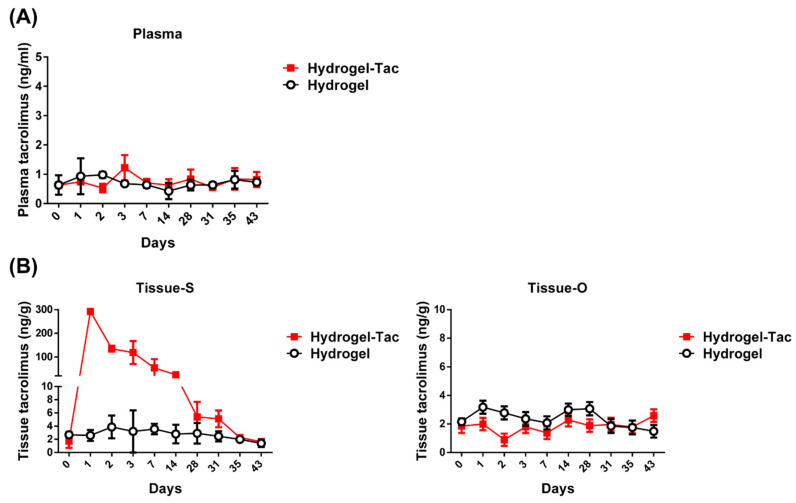
The sustained-release pharmacokinetics of Hydrogel-Tac in the mouse model. 0.1 mL of hydrogel or Hydrogel-Tac was injected into the subcutaneous space of the hindlimb, which was near the transected sciatic nerve. At different time points, plasma and hindlimb muscle tissue (100 mg) were harvested from the mice treated with one subcutaneous injection of 0.1 mL of hydrogel or Hydrogel-Tac (2 mg/mL). 100 mg of muscle tissue was further digested to release tacrolimus and then dissolved in 300 μL of lysis buffer for the tacrolimus ELISA. (**A**) Tacrolimus concentration (ng/mL) in the blood of the hydrogel-injected mice was detected via tacrolimus ELISA. (**B**) Tacrolimus concentration (ng/g) in the hindlimb tissue of the hydrogel-injected mice was analyzed by tacrolimus ELISA. Tissue-S refers to the right hindlimb muscle and sciatic nerve tissue, which were on the same side as the hydrogel injection. Tissue-O refers to the left hindlimb muscle and sciatic nerve tissue, which were on the side opposite to the hydrogel injection. Day 0 means samples harvested from naïve C57BL/6 mice without any hydrogel injection. Each group had three mice. Mean ± SD was used to represent the statistical data.

**Figure 4 pharmaceutics-15-00508-f004:**
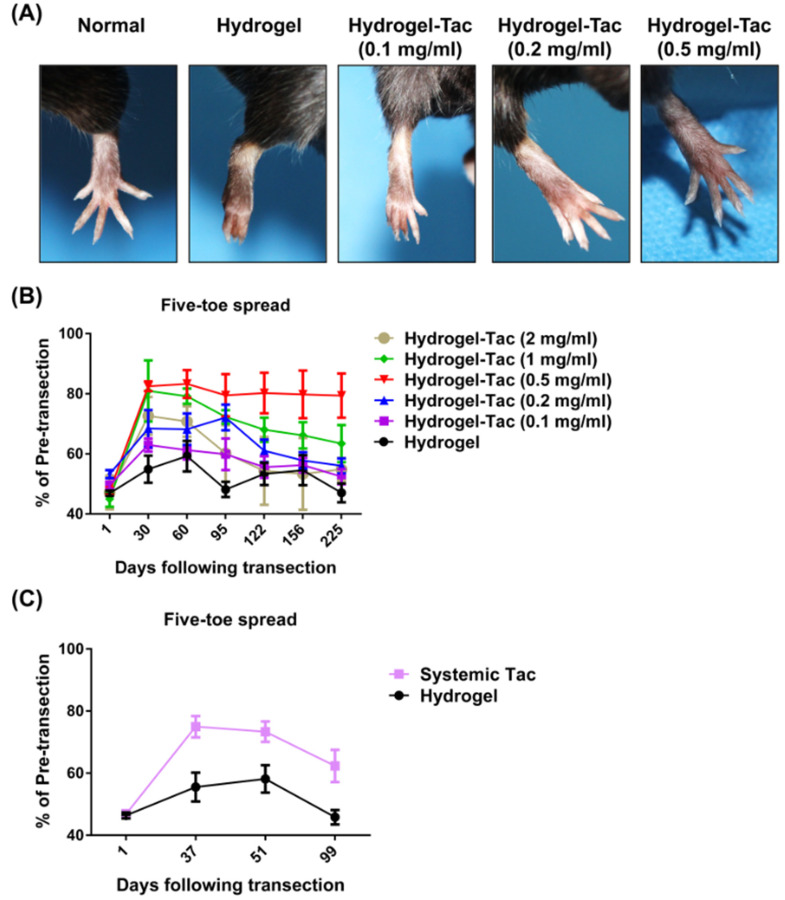
The efficacy of Hydrogel-Tac on the sciatic nerve recovery, assessed using five-toe spread analysis. (**A**) Representative photographs of five-toe spread on postoperative day 94 in different groups. Normal control means that the sciatic nerve was not transected, indicating that the five toes of the feet can normally and completely open. In the hydrogel and Hydrogel-Tac groups, the sciatic nerves of the mice’s right hindlimb were cut and then anastomosed with 10/0 sutures. Subsequently, 0.1 mL of hydrogel or encapsulated-tacrolimus hydrogel at one of various concentrations was injected into the subcutaneous space of the right hindlimb. (**B**) The measurements of the five-toe spread in the hydrogel and Hydrogel-Tac groups following sciatic nerve transection up to postoperative day (POD) 225. The cumulative graph is expressed as a percentage of the postoperative five-toe spread distance divided by the pretransection distance. The hydrogel, Hydrogel-Tac (0.1 mg/mL), Hydrogel-Tac (0.2 mg/mL), Hydrogel-Tac (0.5 mg/mL), Hydrogel-Tac (1 mg/mL), and Hydrogel-Tac (2 mg/mL) groups had six, five, five, eight, five, and six mice, respectively. The statistical comparison of mean ± SD of the hydrogel group with the other groups displayed a significant difference (*p* = 0.0001, one-way ANOVA; hydrogel vs. Hydrogel-Tac (0.1 mg/mL), *p* = 0.0994; hydrogel vs. Hydrogel-Tac (0.2 mg/mL), *p* = 0.0447; hydrogel vs. Hydrogel-Tac (0.5 mg/mL), *p* = 0.0077; hydrogel vs. Hydrogel-Tac (1 mg/mL), *p* = 0.0297; hydrogel vs. Hydrogel-Tac (2 mg/mL), *p* = 0.2345; Tukey’s test). The average distance of the five-toe spread in the normal mouse group was 9.998 ± 0.05317 mm. (**C**) The measurements of the five-toe spread in the hydrogel and systemic Tac groups following sciatic nerve transection up to POD 99. The injection dosage of the hydrogel group was the same as above. Each mouse in the systemic Tac group received an intraperitoneal injection (2 mg/kg), daily, until POD 99. Both groups contained six mice. Mean ± SD was used to represent the statistical data. The comparison of the hydrogel group with the systemic Tac group showed a significant difference on POD 37, 51, and 99 (POD 37, *p* = 0.0071; POD 51, *p* = 0.021; POD 99, *p* = 0.0158, unpaired *t*-test).

**Figure 5 pharmaceutics-15-00508-f005:**
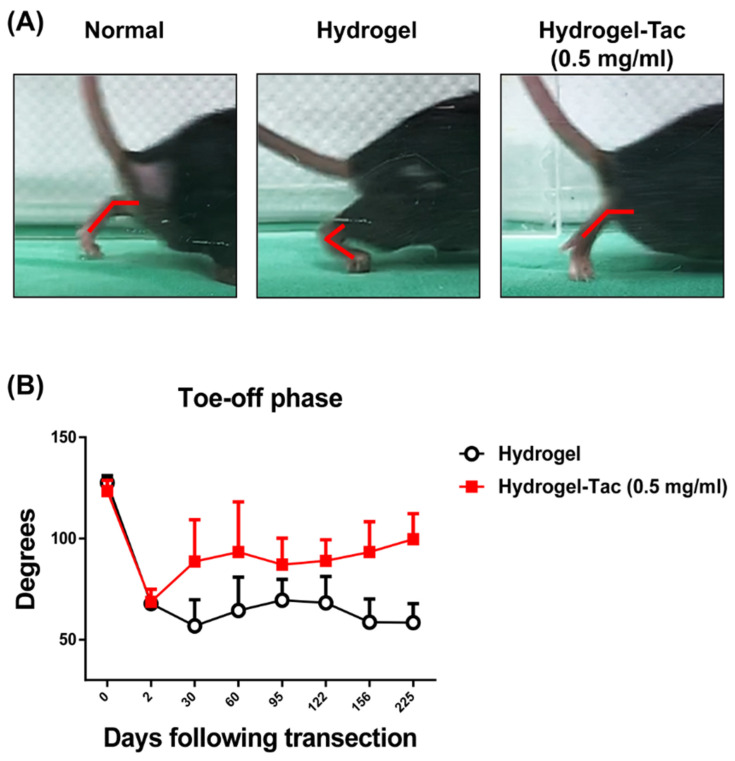
The efficacy of Hydrogel-Tac on sciatic nerve recovery, assessed using video gait analysis. (**A**) Representative photographs of the gait angles on POD 122 in the different groups. Normal control means that the sciatic nerve was not transected, indicating that the gait angle of the feet in the toe-off phase showed a large angle. In the hydrogel and Hydrogel-Tac groups, the sciatic nerves of the right hindlimb were cut and then anastomosed with 10/0 sutures. Subsequently, 0.1 mL of hydrogel or encapsulated-tacrolimus hydrogel (0.5 mg/mL) was injected into the subcutaneous space of the right hindlimb. (**B**) The analysis of the gait angles in the hydrogel and Hydrogel-Tac groups following sciatic nerve transection up to POD 225. The cumulative graph expresses the gait angle of the toe-off phase postoperatively. POD 0 indicates the gait angle measured when the sciatic nerve had not yet been transected. Both groups contained six mice. Mean ± SD was used to represent the statistical data. The comparison of the hydrogel group with the Hydrogel-Tac (0.5 mg/mL) group showed a significant difference on POD 30, 95, 122, 156, and 225 (POD 30 *p* = 0.0095, POD 95 *p* = 0.027, POD 122 *p* = 0.0121, POD 156 *p* = 0.0011, and POD 225 *p* < 0.0001, unpaired *t*-test). The average angle of the normal mouse group was 125.6 ± 1.392.

**Figure 6 pharmaceutics-15-00508-f006:**
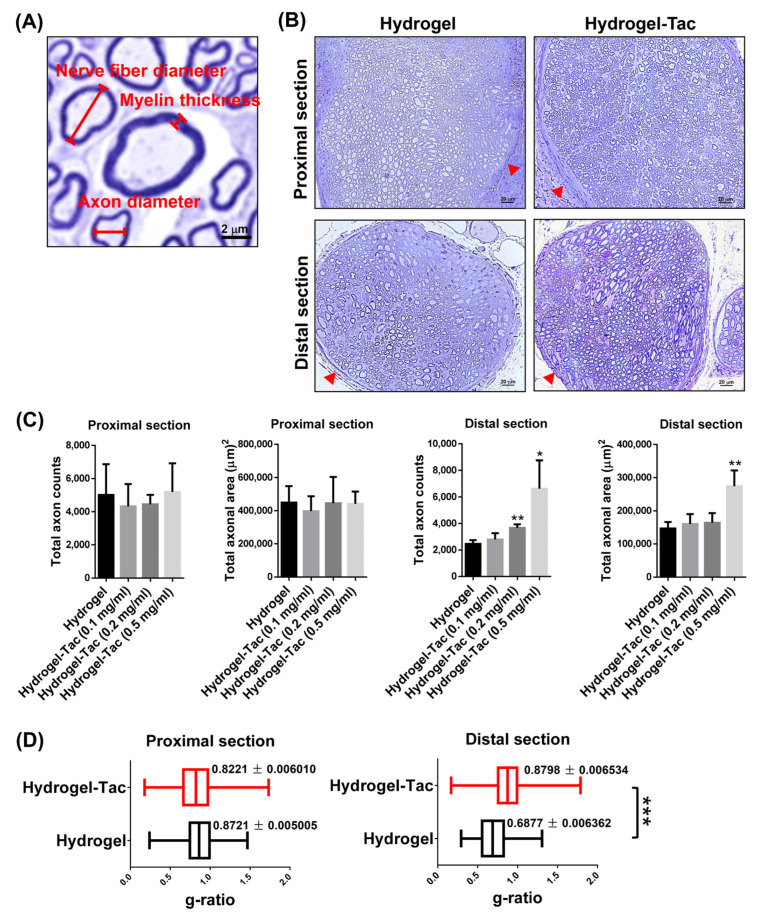
Axonal analysis of proximal and distal sciatic nerve ends between Hydrogel and Hydrogel-Tac groups. In both groups, the sciatic nerves of the right hindlimb were cut and then anastomosed with 10/0 sutures. Subsequently, 0.1 mL of hydrogel or encapsulated-tacrolimus hydrogels (0.5 mg/mL) were injected into the subcutaneous space of the right hindlimb. (**A**) Representative photographs of the toluidine-blue-stained sciatic nerve. The labeled bars represent the myelin thickness, nerve fiber diameter, and axon diameter. The scale bars is 2 μm. (**B**) Microscopic evaluation of the cross-sectional and toluidine-blue-stained sciatic nerves in both groups on POD 225. The perineurium is indicated by the red arrow. The scale bars are 20 μm. (**C**) The total axon counts and total axonal area in the proximal and distal sciatic nerve ends. Each group had three mice. (**D**) The g ratio in the proximal and distal sciatic nerve ends of both groups is expressed as a ratio of the axon diameter divided by the nerve fiber diameter. It reflects the nerve conduction speed. In the proximal and distal sections, both groups contained ratios of at least 1400 and 1010. The total axon counts, total axonal area, and g ratio were calculated using Image-Pro 2D/3D/4D image analysis software. The statistical comparison of mean ± SD of the hydrogel group with Hydrogel-Tac group displayed a significant difference (* *p* < 0.05, ** *p* < 0.05, *** *p* < 0.05, unpaired *t*-test).

**Figure 7 pharmaceutics-15-00508-f007:**
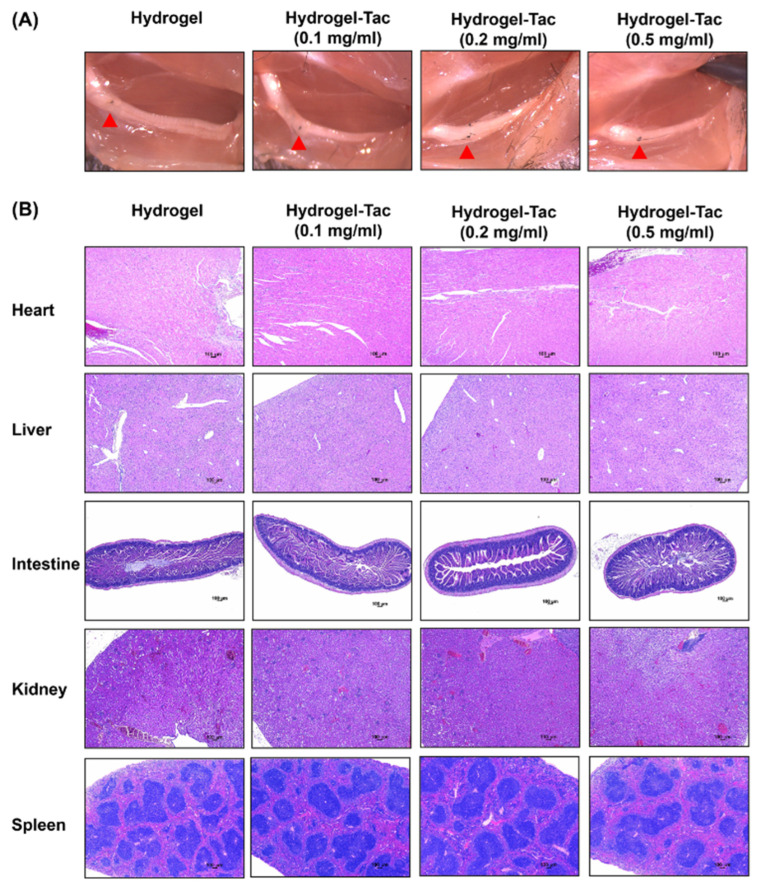
The long-term safety profile of Hydrogel-Tac in mouse transected sciatic nerve model. In the hydrogel and Hydrogel-Tac groups, the sciatic nerves of the mice’s right hindlimb were cut and then anastomosed with 10/0 sutures. Subsequently, 0.1 mL of hydrogel or encapsulated-tacrolimus hydrogel was injected into the subcutaneous space of the right hindlimb. (**A**) Macroscopic examination of the sutured sciatic nerve over 289 days in different groups. The sutured site of surgical repair is indicated by the red arrow. (**B**) Microscopic evaluation of various organs stained with hematoxylin and eosin in different groups. The scale bars are 100 μm.

## Data Availability

The authors confirm that the data supporting the findings of this study are available within the article and its Appendix A. Additionally, the data that support the findings of this study are available from the corresponding author (A.Y.L.W.: aline2355@yahoo.com.tw), upon reasonable request.

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
