# Peer review of "Sustained Release of Tacrolimus Embedded in a Mixed Thermosensitive Hydrogel for Improving Functional Recovery of Injured Peripheral Nerves in Extremities"

_pharmaceutics, 2023, doi:10.3390/pharmaceutics15020508_

Round 1

Reviewer 1 Report

The paper describes a thermosensitive hydrogel system carrying tacrolimus. Experiments were carried out using different in vitro and in vivo assays. The authors demonstrated that the hydrogel drug degraded in a sustained manner and locally released tacrolimus in a small animal model without affecting systemic immunity. Furthermore, the hydrogel drug significantly improved the functional recovery of injured sciatic nerves. Finally, the authors showed that a local injection of an encapsulated-tacrolimus mixed thermosensitive hydrogel accelerated peripheral nerve recovery while having no systemic side effects.

The paper is organized in a logical and clear flow, and the experiments are well-designed and described. The reported conclusions are substantiated by the obtained results, but need comments on the limitations and perspective of this study.

However, the only concern is related to the existing granted patent supporting the same research (granted patent number TWI649096B, https://patents.google.com/patent/TWI649096B/en?oq=TWI649096B).

I am not convinced of the novelty and originality of the data presented in this manuscript.

The authors should emphasize this point in the manuscript and explain the distinction between the two studies.

Author Response

REVIEWER 1

The paper describes a thermosensitive hydrogel system carrying tacrolimus. Experiments were carried out using different in vitro and in vivo assays. The authors demonstrated that the hydrogel drug degraded in a sustained manner and locally released tacrolimus in a small animal model without affecting systemic immunity. Furthermore, the hydrogel drug significantly improved the functional recovery of injured sciatic nerves. Finally, the authors showed that a local injection of an encapsulated-tacrolimus mixed thermosensitive hydrogel accelerated peripheral nerve recovery while having no systemic side effects.

The paper is organized in a logical and clear flow, and the experiments are well-designed and described. The reported conclusions are substantiated by the obtained results, but need comments on the limitations and perspective of this study.

However, the only concern is related to the existing granted patent supporting the same research (granted patent number TWI649096B, https://patents.google.com/patent/TWI649096B/en?oq=TWI649096B).

I am not convinced of the novelty and originality of the data presented in this manuscript.

The authors should emphasize this point in the manuscript and explain the distinction between the two studies.

Ans: Thank you very much for your comments. Actually, this TWI649096B is our patent (our coauthor's patent: Prof. I-Ming Chu, please see the authorship). This patent description is mentioned in the patent section of the manuscript and is cited in reference 55 of the manuscript. Our reference also showed the patent number below:

  1. I-Ming Chu, H.-C.L., Chih-Chi Cheng, Kuan-Lin Ku, Drug carrier and drug delivery system using the same. Taiwan Patent Search System, 2019: p. Patent number: I649096.

This patent can be searched out in the "Taiwan Patent Search System" using the patent number "I649096". TW means Taiwan. B means patent announcement (gazette). To make the patent description clear, we have added the patent number TWI649096B to the patent section of the manuscript. Detail infpormation can be seen Figures below.

In order to develop an innovative hybrid hydrogel, which allows a sustained release of drugs to accelerate peripheral nerve regeneration in extremities, our CGMH (Chang Gung Memorial Hospital, Taiwan) VCA team cooperated with Prof. I-Ming Chu (National Tsing Hua University, Taiwan). Therefore, Prof. Chu's team developed new extended-release delivery systems of tacrolimus using new hydrogel materials and then provided us with this Hydrogel-Tac to further understand the potential of this hydrogel bioconstruct on the sustained release of FK506 in order to accelerate peripheral nerve regeneration. This research was funded by a three-year integrated grant (2018-2021) from the Ministry of Science and Technology, Taiwan. Prof. Chu performed sub-grant I to develop the new hybrid hydrogel and the title of the sub-grant I is "Integration of Innovative Hybrid Hydrogel to Improve Immunosuppressant Drug Delivery System and Optimize the Outcome of Long-term Allotransplantation". We performed the sub-grant III and its title is "Investigating the effect of a sustained release of Tacrolimus embedded in a novel bioengineered hydrogel in improving functional recovery of injured and repaired peripheral nerve in extremities" and the grant number of the third year is " MOST 109-2314-B-182-074".

The synthesis and characterization of this hydrogel was published in "Pharmaceutics. 2019 Aug 14;11(8):413.", and paper title is "A Mixed Thermosensitive Hydrogel System for Sustained Delivery of Tacrolimus for Immunosuppressive Therapy". This paper was mentioned in the manuscript throughout.

Therefore, the effects of Hydrogel-Tac using the mixed thermosensitive hydrogel system on the regeneration of injured peripheral nerves was first investigated in this study.

If our explanation or description is still unclear, please feel free to let us know.

REVIEWER 2

The author reported the use of thermo-sensitive hydrogel for improving functional recovery of injured peripheral nerves. This is well written manuscript, and the presentation of the result is clear. There are however quite a few points that require the authors attention before the paper can be accepted for publication. Some comments and suggestions are listed as follows:

  1. Why this material is so important in-terms drug delivery vehicle compared to the reported other thermo-responsive gel?

Ans: Thank you very much for your comments.

Because tacrolimus is a hydrophobic substance with low aqueous solubility. Controlled delivery of tacrolimus would avoid the low bioavailability problem and improve patient compliance statistics by reducing the administration frequency. Several delivery systems for tacrolimus have been reported, including mPEG-poly(lactic acid) nanoparticles [1], poly(lactide-co-glycolide) microspheres [2], and triglycerol monostearate hydrogels [3]. The in situ gelation property of the system studied here has several advantages over particulate systems or conventional hydrogel systems, including longer release time, higher encapsulation efficiency, easy administration, and applicability to various types of drugs. In particular, by combining two types of hydrogel components, the possibility of increasing the flexibility of carriers to suit distinct pharmaceutical entities and modifying release rates for better clinical outcomes is explored.

This description was from our previous paper (A Mixed Thermosensitive Hydrogel System for Sustained Delivery of Tacrolimus for Immunosuppressive Therapy. Pharmaceutics. 2019 Aug 14;11(8):413.)

  1. Xu, W.; Ling, P.; Zhang, T. Toward immunosuppressive effects on liver transplantation in rat model: Tacrolimus loaded poly (ethylene glycol)-poly (d,l-lactide) nanoparticle with longer survival time. Int. J. Pharm. 2014, 460, 173–180.
  2. Tajdaran, K.; Shoichet, M.S.; Gordon, T.; Borschel, G.H. A novel polymeric drug delivery system for localized and sustained release of tacrolimus (FK506). Biotechnol. Bioeng. 2015, 112, 1948–1953.
  3. Gajanayake, T.; Olariu, R.; Leclere, F.M.; Dhayani, A.; Yang, Z.; Bongoni, A.K.; Banz, Y.; Constantinescu, M.A.; Karp, J.M.; Vemula, P.K.; et al. A single localized dose of enzyme-responsive hydrogel improves long-term survival of a vascularized composite allograft. Sci. Transl. Med. 2014, 6, 249ra110.

  1. In line 95, author mentioned molar mass of the polymer is 1.5. I think this is dispersity of the polymer.

Ans: Thank you very much for your comments. Apology for this. We have changed it to “the molar mass polydispersity is 1.51.”

  1. What is the reason to use mixed hydrogel? Why not only Pluronic F-127 or P–Lys–Ala–PLX separately?

Ans: Thank you very much for your comments. We described the reason for the use of the mixed hydrogel composed of P–Lys–Ala–PLX and Pluronic F-127 in our previous paper (A Mixed Thermosensitive Hydrogel System for Sustained Delivery of Tacrolimus for Immunosuppressive Therapy. Pharmaceutics. 2019 Aug 14;11(8):413.) Please see the information below:

3.2. Development of Mixed Hydrogel Formulation

After the preliminary in vivo drug release tests with the P–Lys–Ala–PLX hydrogel, extremely slow tacrolimus release rates and a considerably low plasma concentration of tacrolimus were observed. Unexpected transplant rejection resulted from these tests (data not shown) because drug concentrations were low. The slow in vivo release rate of the drug may be caused by the slow degradation rate of the hydrogel in vivo. Therefore, the hydrogel formulation should be modified to accelerate drug release at effective concentrations and sustain the release for more than 30 days. Pluronic F-127 has been mentioned to modulate the drug release rate [1]. The addition of Pluronic F-127, an approved fast-degrading hydrogel, to the P–Lys–Ala–PLX hydrogel system may provide a solution to accelerate the release rate.

A series of hydrogel formulations of 1 to 3 wt % Pluronic F-127 powder mixed into 4 or 5 wt % P–Lys–Ala–PLX were studied, as shown in Table 2. Sol-to-gel transition properties, cytotoxicity, drug encapsulation efficiency, and the release rate were measured. The formulation of 5 wt % P–Lys–Ala–PLX with 1 wt % Pluronic F-127 (sample 5:1) had a lower transition temperature than the other three groups, as demonstrated in Figure 4A. Additionally, the sample 5:1 had the highest drug encapsulation efficiency (Figure 4B). Consequently, the formulation of the sample 5:1 was chosen for further studies.

  1. Kojarunchitt, T.; Hook, S.; Rizwan, S.; Rades, T.; Baldursdottir, S. Development and characterisation of modified poloxamer 407 thermoresponsive depot systems containing cubosomes. Int. J. Pharm. 2011, 408, 20–26.

  1. The author used thermoresponsive hydrogel as a drug delivery vehicle. But it is not clear for the reader why the drug is releasing at 37 oC.

Ans: Thank you very much for your comments. We have added several sentences in the introduction using the yellow highlights for more clarity.

A thermosensitive hydrogel has a hydrophobic group in its structure in addition to the hydrophilic group. Both hydrophilic and hydrophobic groups compete with each other. When the temperature is low, hydrogen bonding occurs between water molecules and hydrophilic groups. Thus, the polymer chain can be completely stretched in water after absorbing water to form a uniform solution. When a certain temperature is reached, the hydrogen bonding force between molecules becomes weaker. Thus, the hydrophobic group aggregates in water and makes the polymer solution change from a liquid state to a colloidal state.

We found that Pluronic F-127 help the mixed hydrogel to release the drug through the acceleration of hydrogel degradation at 37 °C compared with P–Lys–Ala–PLX hydrogel only. (This result was from our previous paper (A Mixed Thermosensitive Hydrogel System for Sustained Delivery of Tacrolimus for Immunosuppressive Therapy. Pharmaceutics. 2019 Aug 14;11(8):413.)

If our description is still unclear, please feel free to let us know.

Reviewer 2 Report

The author reported the use of thermo-sensitive hydrogel for improving functional recovery of injured peripheral nerves. This is well written manuscript, and the presentation of the result is clear. There are however quite a few points that require the authors attention before the paper can be accepted for publication. Some comments and suggestions are listed as follows:

1.      Why this material is so important in-terms drug delivery vehicle compared to the reported other thermo-responsive gel?

2.     In line 95, author mentioned molar mass of the polymer is 1.5. I think this is dispersity of the polymer.

3.     What is the reason to use mixed hydrogel? Why not only Pluronic F-127 or P–Lys–Ala–PLX separately?

4.     The author used thermoresponsive hydrogel as a drug delivery vehicle. But it is not clear for the reader why the drug is releasing at 37 oC.

Author Response

(The authors gave the same response as above.)

Round 2

Reviewer 1 Report

Accept in present form